# Blood Bacterial DNA Load and Profiling Differ in Colorectal Cancer Patients Compared to Tumor-Free Controls

**DOI:** 10.3390/cancers13246363

**Published:** 2021-12-18

**Authors:** Massimiliano Mutignani, Roberto Penagini, Giorgio Gargari, Simone Guglielmetti, Marcello Cintolo, Aldo Airoldi, Pierfrancesco Leone, Pietro Carnevali, Clorinda Ciafardini, Giulio Petrocelli, Federica Mascaretti, Barbara Oreggia, Lorenzo Dioscoridi, Federica Cavalcoli, Massimo Primignani, Francesco Pugliese, Paola Bertuccio, Pietro Soru, Carmelo Magistro, Giovanni Ferrari, Michela C. Speciani, Giulia Bonato, Marta Bini, Paolo Cantù, Flavio Caprioli, Marcello Vangeli, Edoardo Forti, Stefano Mazza, Giulia Tosetti, Rossella Bonzi, Maurizio Vecchi, Carlo La Vecchia, Marta Rossi

**Affiliations:** 1Digestive and Interventional Endoscopy Unit, ASST Grande Ospedale Metropolitano Niguarda, 20162 Milan, Italy; massimiliano.mutignani@ospedaleniguarda.it (M.M.); marcello.cintolo@ospedaleniguarda.it (M.C.); giulio.petrocelli@ospedaleniguarda.it (G.P.); lorenzo.dioscoridi@ospedaleniguarda.it (L.D.); francesco.pugliese@ospedaleniguarda.it (F.P.); giulia.bonato@ospedaleniguarda.it (G.B.); marta.bini@ospedaleniguarda.it (M.B.); edoardo.forti@ospedaleniguarda.it (E.F.); 2Gastroenterology and Endoscopy Unit, Foundation IRCCS Ca’ Granda Ospedale Maggiore Policlinico, 20122 Milan, Italy; roberto.penagini@unimi.it (R.P.); clorinda.ciafardini@policlinico.mi.it (C.C.); federica.mascaretti@unimi.it (F.M.); paolo.cantu@policlinico.mi.it (P.C.); flavio.caprioli@unimi.it (F.C.); stefano.mazza@asst-cremona.it (S.M.); maurizio.vecchi@unimi.it (M.V.); 3Department of Pathophysiology and Transplantation, University of Milan, 20122 Milan, Italy; 4Department of Food, Environmental and Nutritional Sciences (DeFENS), University of Milan, 20133 Milan, Italy; giorgio.gargari@unimi.it (G.G.); simone.guglielmetti@unimi.it (S.G.); 5Hepatology and Gastroenterology Unit, ASST Grande Ospedale Metropolitano Niguarda, 20162 Milan, Italy; aldo.airoldi@ospedaleniguarda.it (A.A.); marcello.vangeli@ospedaleniguarda.it (M.V.); 6General Surgery Unit, Foundation IRCCS Ca’ Granda Ospedale Maggiore Policlinico, 20122 Milan, Italy; pierfrances.leone@policlinico.mi.it (P.L.); barbara.oreggia@policlinico.mi.it (B.O.); 7Division of Minimally-Invasive Surgical Oncology, Niguarda Cancer Center, ASST Grande Ospedale Metropolitano Niguarda, 20162 Milan, Italy; pietro.carnevali@ospedaleniguarda.it (P.C.); carmelo.magistro@ospedaleniguarda.it (C.M.); giovanni.ferrari@ospedaleniguarda.it (G.F.); 8Associazione Nazionale Operatori Tecniche Endoscopiche (ANOTE), 80061 Massa Lubrense, Italy; 9Department of Clinical Science and Community Health, University of Milan, 20133 Milan, Italy; paola.bertuccio@unipv.it (P.B.); michela.speciani@unimi.it (M.C.S.); rossella.bonzi@unimi.it (R.B.); carlo.lavecchia@unimi.it (C.L.V.); 10Diagnostic and Therapeutic Endoscopy Unit, Fondazione IRCCS Istituto Nazionale Tumori, 20133 Milan, Italy; federica.cavalcoli@istitutotumori.mi.it; 11Division of Gastroenterology and Hepatology, Foundation IRCCS Ca’ Granda Ospedale Maggiore Policlinico, 20122 Milan, Italy; massimo.primignani@policlinico.mi.it (M.P.); giulia.tosetti@policlinico.mi.it (G.T.); 12Department of Public Health, Experimental and Forensic Medicine, University of Pavia, 27100 Pavia, Italy; 13Division of Endoscopy, IRCCS Istituto Europeo di Oncologia, 20141 Milan, Italy; pietro.soru@ieo.it; 14Gastroenterology and Digestive Endoscopy Unit, ASST Cremona, Cremona 26100, Italy

**Keywords:** bacterial translocation, bacterial 16S rRNA gene, case-control study, colon cancer diagnoses, microbiome

## Abstract

**Simple Summary:**

In colorectal cancer patients, epithelial barrier dysfunction can lead to increased intestinal permeability, and gut microbiome was found to vary compared to healthy subjects. We conducted a study to investigate whether bacterial translocation from gastrointestinal tract to bloodstream is associated to intestinal adenoma and/or colorectal cancer. In particular, an epidemiological and metagenomic approach was used to evaluate the relation of the bacterial DNA load and the bacterial taxonomic groups—assessed by 16S rRNA profiling—in blood with the risks of intestinal adenoma and colorectal cancer. These findings can confirm the presence of bacterial DNA in blood in healthy adults and serve as a basis to evaluate new non-invasive techniques for an early CRC diagnosis through the analyses of bacterial DNA circulating in peripheral blood.

**Abstract:**

Inflammation and immunity are linked to intestinal adenoma (IA) and colorectal cancer (CRC) development. The gut microbiota is associated with CRC risk. Epithelial barrier dysfunction can occur, possibly leading to increased intestinal permeability in CRC patients. We conducted a case-control study including 100 incident histologically confirmed CRC cases, and 100 IA and 100 healthy subjects, matched to cases by center, sex and age. We performed 16S rRNA gene analysis of blood and applied conditional logistic regression. Further analyses were based on negative binomial distribution normalization and Random Forest algorithm. We found an overrepresentation of blood 16S rRNA gene copies in colon cancer as compared to tumor-free controls. For high levels of gene copies, community diversity was higher in colon cancer cases than controls. Bacterial taxa and operational taxonomic unit abundances were different between groups and were able to predict CRC with an accuracy of 0.70. Our data support the hypothesis of a higher passage of bacteria from gastrointestinal tract to bloodstream in colon cancer. This result can be applied on non-invasive diagnostic tests for colon cancer control.

## 1. Introduction

Colorectal cancer (CRC) is the 3rd most common cancer, and ranks second in terms of mortality, worldwide [1]. Although mortality trends have been favorable in Europe during the last decades, with a predicted rate of 15.4/100,000 in men and 8.6/100,000 in women in 2020, in most eastern European countries CRC mortality is still increasing [2,3].

CRC derives from a sequential accumulation of genetic alterations that involves the transition from normal mucosa to pre-malignant lesions with progression to intestinal adenoma (IA) and invasive CRC [4]. Inflammation and immunity are inextricably linked to all phases of CRC development [5]. Numerous studies identified chronic intestinal inflammation as a risk factor for CRC, as also confirmed by an increased incidence of this tumor in inflammatory bowel disease patients [6]. IA and CRC have also been associated with increased circulating inflammation [7,8], and more recently with dysfunction of the gut mucosal barrier [9].

The community structure of the intestinal microbial ecosystem influences the risk of IA and CRC [9,10,11]. Gut mucosal microbiota *Fusobacterium* spp. were increased in CRC patients [10,11]; *Bacteroides fragilis* and the genus *Porphyromonas* have been also associated with an increased risk of CRC [10]. Numerous studies analyzed fecal microbiome, not always reporting consistent results [11,12,13], but recent meta-analyses including geographically and technically different shotgun metagenomic studies showed a higher fecal microbiota richness in CRC cases as compared to controls [14], and identified a set of bacterial taxa significantly enriched in CRC cases [14,15].

Increasing evidence has pointed out the presence of bacterial DNA in blood [16,17]. Microbial signatures have been reported for gut dysbiosis among diabetic subjects and for liver fibrosis in obese patients [18,19]. Microbiome analysis of blood has also been proposed as a tool to discriminate between cancer patients and healthy subjects [20]. In particular, differences in circulating bacterial factors can occur in CRC patients, in whom epithelial barrier dysfunction can lead to increased intestinal permeability, plausibly resulting into a greater bacterial translocation from the gastrointestinal tract to bloodstream in IA and/or CRC. Some differences in relative abundance of the bacterial DNA in plasma between healthy, IA and CRC subjects have been reported in a case-control study from China, including 57 participants, but differences in terms of total bacterial load have not been analyzed yet [21].

In this study, we aimed to compare the load of bacterial DNA in blood and taxonomic profile between CRC, IA and healthy controls, using data from a case-control study.

## 2. Results

Table 1 gives the distribution of 100 healthy controls, 100 IA subjects, and 100 cases of CRC according to sex, age, study center and education. By design, the three groups had the same sex and center distributions and were similar in terms of age. Cases tended to be less educated than IA subjects and controls in absence, however, of a significant difference (χ^2^ test *p* = 0.155).

### 2.1. 16S rRNA Gene Copies

We found an overall mean of 7687 16S rRNA gene copies per µL of blood, with a mean of 7628 among controls, 7586 among IA and 8387 among CRC subjects (9145 in colon and 7629 in rectal cancers), with no significant differences between the three groups (*p* for heterogeneity = 0.482) (Appendix A). Since 16S rRNA gene copy distribution was very similar in control and IA subjects (*p* for heterogeneity = 0.95), we grouped them as reference group and compared their 16S rRNA gene copy distribution with that of CRC, colon and rectal cancer cases. We found significant differences between colon cancer and control/IA (*p* = 0.025; Figure 1).

Table 2 shows the distribution of control/IA subjects, CRC, colon and rectal cancer cases, the odds ratios (ORs) and the corresponding 95% confidence intervals (Cis) according to quintiles of 16S rRNA gene copies, as well as the continuous ORs. We found a direct association of 16S rRNA gene copies with the risk of colon cancer r. The OR of colon cancer for the highest (≥9707) versus the lowest first three (<7618) quintiles of gene copies was 2.62 (95%CI = 1.22–5.65). The association significantly increased after the fourth quintile (*p* for trend = 0.013) and became stronger for levels higher than the fifth quintile cut-off: the OR was 7.22 (95%CI = 2.18–23.9) after the 90th centile (>11,265 copies) and 17.08 (95%CI = 3.36–86.87) after the 95th centile (>13,000 copies). The continuous OR indicated a two-fold increased risk for an increment equal to 4328 copies (OR = 2.02; 95%CI = 1.26–3.25). No association was found between 16S rRNA gene copies and rectal cancer risk. The OR for the highest versus the lowest three quintiles was 0.81 (95%CI = 0.32–2.03). The test for heterogeneity between colon and rectal results was significant (*p* = 0.021).

When the association with 16S rRNA gene copies was further examined according to colon subsites, a positive association appeared to be more pronounced for right colon, with an OR for the highest versus the lowest three quintiles of 10.75 (95%CI = 2.16–53.42) as compared to 1.25 (95%CI = 0.46–3.38) for other colon sites, in absence, however, of heterogeneity (*p* = 0.123). Among 21 right colon cancers, 11 (53%) were in the highest quintile of gene copies, whereas among 29 other colon cancer subsites, 8 (26%) were in the highest quintile, in comparison to 40 out of 200 (20%) among control/IA subjects. In particular, among 11 cancers of the ascending colon, 7 (64%) were in the highest quintile (Table 3).

### 2.2. Alpha and Beta Diversity

No differences between groups were found in terms of α-diversity indices (Appendix A). When we restricted the analyses to subjects in the two highest quintiles of 16S rRNA gene copies, we found a higher diversity in colon cancer cases as compared to controls in terms of observed taxa and Chao1 indices for both genera (median of 32 vs. 28, *p* = 0.054 and median of 49 vs. 40.6, *p* = 0.059, respectively) and operational taxonomic units (OTUs) (median of 40 vs. 34, *p* = 0.039 and median of 71.1 vs. 53.4, *p* = 0.067), and as compared to rectal cancer cases in terms of both observed genera and OTUs (median of 32 vs. 29, *p* = 0.023 and median of 40 vs. 37, *p* = 0.029) (Table 4, Figure 2). Colon cancers also appeared to be higher than IA in terms of observed genera (median of 32 vs. 28, *p* = 0.071), and when we compared the observed genera index in colon cancer cases versus control/IA subjects together, the *p* value for heterogeneity decreased to 0.035.

Concerning the beta diversity, no differences between groups were found overall and among subjects into the two highest quintiles of 16S rRNA gene copies. When we restricted the analyses to the highest quintile of 16S rRNA, we observed significant differences between control/IA, colon and rectal cancer patients (Weighted UniFrac, *p* = 0.026; Unweighted UniFrac, *p* = 0.051; Generalized UniFrac, *p* = 0.031) (Figure 3A–C). Post-hoc analyses splitting the groups 2 by 2 showed a trend between controls/IA and colon cancers in Weighted UniFrac (*p* = 0.073) (Figure 3D).

### 2.3. Taxonomic Profiling of Blood Bacterial DNA between Groups

We detected a total of 1081 OTUs that were taxonomically classified into 15 phyla, 34 classes, 87 orders, 164 families and 325 genera.

Pseudomonadaceae, Micrococcaceae, Burkholderiaceae, Caulobacteraceae, Moraxellaceae and Flavobacteriaceae were the six most represented families, which together accounted for more than 50% of all reads assigned to bacterial taxa (Appendix A).

The mean of DESeq2 normalized data and the adjusted *p* values comparing CRC versus control/IA on all the taxonomy levels and OTUs are shown in Figure 4. CRC samples were characterized by the increase of sequencing reads assigned to the bacterial families Peptostreptococcaceae and Acetobacteriaceae, together with a lower representation of the bacterial families Bacteroidaceae, Lachnospiraceae, and Ruminococcaceae. Comparisons between every two groups (CRC versus control, CRC versus IA and IA versus control) are also shown in Appendix A.

Through the Random Forest supervised method, we found a set of variables that predict a CRC case from control/IA subjects with an accuracy of 0.70 (Sensitivity = 0.45; Specificity = 0.87) and another model that inferred the location of CRC discriminating colon from rectal cancer with an accuracy of 0.77 (Sensitivity = 0.71; Specificity = 0.82) (Figure 5). The first model inferred that the families Acetobacteraceae, Peptostreptococcaceae and Oligoflexaceae, the genus *Melittangium*, and the OTUs belonging to the genera *Acinetobacter*, *Pelomonas*, *Novosphingobium* and *Pajaroellobacter* were the most important variables to predict between CRC or control/IA group (Figure 5A). The biplot in Figure 5A shows a separation between CRC and control/IA subjects due to a higher dispersion of CRC cases. The second model inferred that the families *Peptosteptococcaceae*, *Streptococcaceae* and *Ruminococcaceae*, the genera *Arthrobacter*, *Clostridium* sensu stricto and *Kocuria*, and the OTUs belonging to the genera *Legionella*, *Kocuria* and *Lepisosteus oculatus* were the most important variables to predict the group between colon and rectal cancer, together with other variables that contributed to increase the accuracy of the model, including 16S rRNA gene copies, the phylum *Proteobacteria*, the order *Rhizobiales* and the genus *Bacteroides* (Figure 5B).

When the Random Forest algorithm was applied to the sample belonging to the highest quintile of 16S rRNA gene copies, the accuracy to predict between CRC and control/IA group increased to 0.79.

## 3. Discussion

This study shows that colon cancer patients have an overrepresentation of bacterial DNA in blood as compared to tumor-free subjects, including IA or healthy controls. This result appeared stronger for cancers in right colon, whereas no difference in terms of bacterial load was found for rectal cancer. For high levels of 16S rRNA gene copies (>7618), colon cancers had increased community diversity but did not differ on community evenness from controls.

To our knowledge, no other study conducted an ad hoc data collection to systematically investigate blood bacterial DNA load and profiling in relation to the risk of CRC and/or IA to date. In a Chinese study, circulating bacterial DNA of 25 CRC, 10 IA and 22 healthy subjects was analyzed through whole genome sequencing techniques on plasma samples [21], suggesting that the *Flavobacterium* DNA relative abundance was reduced in CRC and IA (<1%) as compared to control subjects (9.4%); on the contrary, there was a 10-fold increase DNA abundance of genus *Ruminococcus* in CRC (0.2%) as compared to controls (0.02%). In the same study, various attempts to identify bacterial biomarkers of CRC or IA through Random Forest algorithms were proposed, reporting a set of 28 species as important features to discriminate between CRC/IA group and controls, but results were based on small samples (23 and 34 subjects for discovery and validation cohort, respectively). Messaritakis et al. used PCR for the amplification of genomic DNA on blood in order to compare 397 adjuvant or metastatic CRC patients with 32 healthy controls in terms of the presence of 3 bacterial genes [22]. Higher rates of the glutamine synthase gene of *Bacteroides fragilis* and the 5.8S rRNA of *Candida albicans* were observed in CRC patients (*p* < 0.001), especially in metastatic disease. No association was found for the genus *E. coli*.

In our data, CRC patients had an enrichment of Peptostreptococcaceae and Acetobacteriaceae and a reduction of Bacteroidaceae, Lachnospiraceae, and Ruminococcaceae. The latter families are the most represented in the fecal and intestinal microbiota [23], while Peptostreptococcaceae and Acetobacteriaceae are less represented in the human intestine. These two families were also found to be more abundant in chronic kidney disease (CKD) than healthy subjects in a case-control study on blood microbiome including 20 CKD cases and 20 controls [24]. In a study involving 99 CRC cases and 103 controls, high abundance of Lachnospiraceae was negatively correlated with colonic colonisation by oral bacteria, including oral pathogens associated with CRC, suggesting a protective role of Lachnospiraceae, potentially influenced by Western dietary patterns [25]. Ruminococcaceae abundance was found to decrease in CRC tissue as compared to tumor-adjacent biopsies and stool samples from the same case in a study including 294 subjects [26]. Moreover, the members of the families Lachnospiraceae and Ruminococcaceae were recognized as the most active members of the human colonic microbiota [27], able to efficiently convert fibers into butyrate [28], a bacterial catabolite widely demonstrated to regulate T-reg lymphocyte priming preventing CRC [29,30].

Gut microbiota, inflammation and nutrition play an important role on intestinal permeability which may influence the risk of CRC [31,32]. Bacteria have shown capacity to interact directly with immune system cells and to impact in multiple host functions [33,34], but it is unclear which one comes first between local inflammation, intestinally permeability and changes in resident microbiota [35,36]. In this context, it has also been shown that bacteria can disseminate to liver through the disruption of gut vascular barrier in colorectal cancer with hepatic metastasis [37].

Our data corroborate the hypothesis of a greater bacterial translocation from gastrointestinal tract to bloodstream in colon cancer, especially in right colon cancer, but not in rectal cancer patients. Although this result should be interpreted with caution given the reduced number of cases, various etiological factors and biological aspects are different according to CRC sites. Physical activity, antibiotic use and family history of CRC are relevant in colon but not in rectal cancer [37,38,39,40], and associations of some dietary components are stronger for some colon subsites cancers [41,42]. Moreover, the proximal and distal colon have a different embryological origin, also resulting in a distinct vascular supply [43]. Local differences in underlying genetics, including genetic expression and immunological activity have been highlighted [44], along with a negative gradient of immune cells from proximal colon to distal colon and rectum [43]. These features can play a role in changing the extent of bacterial translocation along the lower intestinal tract, leading to differences in blood 16S rRNA gene copies according to CRC localization. Moreover, gut microbiota and mechanisms of carcinogenesis also vary from left to right colon and to rectum [40,45]. Another important factor that could affect regional differences in bacterial translocation and blood 16S rRNA gene copies is the organization and thickness of mucus, which appears to be looser and less organized in the proximal colon, facilitating bacterial contact with the epithelium and, thus, translocation [46,47]. Mucus layers were found to vary along the colon in mice in terms of O-glycosylated entities of Muc2 and of the host-microbiota symbiosis regulation [48].

Recent meta-analyses on fecal microbiome suggested universal, validated predictive taxonomic and functional microbiome CRC signatures, revealing potential mechanisms behind the intestinal carcinogenesis processes, and putting the basis for non-invasive CRC diagnosis through metagenomic analysis of fecal microbiome [14,15]. CRC screening programs can suffer from limited sensitivity and specificity of the tests used and from possible low adherence with CRC screening recommendations—mainly due to the refusal of fecal test and colonoscopy—in some countries [49]. Innovative, non-invasive diagnostic tests would be of support for CRC control [49]. Various tests based on blood analyses have been proposed, including techniques estimating the presence of *Streptococcus bovis* [50] or evaluating the antibody level against *Fusobacterium nucleatum* [51] in blood, and, more recently, the use of blood microbiome profile has been suggested [20].

Our data showed various taxa associated with CRC and identified a set of covariates of taxa and OTUs that is able to predict CRC from control/IA subjects and colon from rectal cancers with an accuracy around 0.7. When restricting the analyses to subjects with high levels of 16S rRNA gene copies, we found more accurate models, but we were not able to separate analysis for colon cancer only and results should be interpreted with caution given the small numbers. However, these findings can serve as a basis to conceive new non-invasive techniques for an early diagnosis of CRC based on bacterial DNA circulating in peripheral blood. In particular, they can be relevant for the detection of right colon cancers, which often have a subtle presentation and a more advanced stage at diagnosis, partly because right colon is more difficult to be explored, when compared to rectal and distal colon [52].

Our data show a non-trivial dysfunction of the intestinal epithelial barrier permeability also for patients without cancer, in line with previous studies that characterized blood microbiota among healthy subjects [17]. It is not easy to explain the lack of difference between rectal cancer and tumor-free subjects, since the bacterial translocation can be different due to the related inflammatory condition in rectal cancer patients [53]. A possible role of size or shape selectivity for bacteria transiting from the intestine to the bloodstream in different locations could explain at least in part this result but a limitation of this study is the lack of fecal samples on our subjects, which prevented us from further analyzing this issue. One of the strengths of this study is the conduction of an ad-hoc data collection, which includes a new developed standardized protocol, fully observed by the recruitment centers. CRC and the corresponding IA and control subjects were comparable in terms of setting since they derived from the same catchment area and recruitment procedures. Moreover, interviewers and investigators were blinded to the group assignment, as data collection was performed before endoscopy and diagnosis. Most cases were detected at the first CRC-diagnosing colonoscopy, allowing us to recruit truly incident cases, characterized by a minimal time between participant’s recruitment and cancer diagnosis and by available clinical data from the very beginning of the diagnostic process. Moreover, the presence of healthy controls allowed a clean comparison with CRC and IA, and the inclusion of IA allowed to investigate an important phase on the mechanisms behind the process of the adenoma-carcinoma sequence. However, the lack of an ad hoc developed validation cohort remains a limitation of our work. We were able to adjust for study center, sex and age, eliminating possible confounding effects of these covariates on 16s rRNA gene copies results.

In conclusion, our data confirm the presence of bacterial DNA in blood in healthy adults and indicate that colon cancer patients had a higher DNA bacterial load and a different bacterial profiling as compared to healthy, IA and rectal cancer subjects, revealing a higher passage of bacteria from gastrointestinal tract to bloodstream in colon cancer. Further studies are needed to confirm this result and possibly exploit it for the development of innovative early techniques for colon cancer diagnosis.

## 4. Materials and Methods

We conducted an ad hoc developed case-control study between May 2017 and November 2019 in the metropolitan area of Milan, Italy. Recruitment centers included two general hospitals of Milan: the Digestive and Interventional Endoscopy Unit, Azienda Socio Sanitaria Territoriale (ASST) Grande Ospedale Metropolitano Niguarda, the coordinator center, and the Gastroenterology and Endoscopy Unit, Fondazione Istituto di Ricovero e Cura a Carattere Scientifico (IRCCS) Ca’ Granda Ospedale Maggiore Policlinico. Both hospitals included a colonoscopy screening referral center of the CRC screening program, managed by Health Protection Agency.

CRC cases were enrolled together with non-cancer adenomatous polyps and healthy controls, frequency-matched with cases by center, age ± 5 years and sex. Trained interviewers recruited study participants among eligible outpatients or inpatients who were scheduled for a colonoscopy, including patients referred for the CRC screening program. Excluded criteria were: (1) colonoscopy in the last 5 days; (2) reported previous cancers; (3) inflammatory chronic bowel diseases, (4) liver or kidney failure (creatinine ≥ 1.7 mg/dL, dialysis); (5) NYHA grade III or IV heart failure; (6) primary or secondary immunodeficiency; (7) recent hospitalization (1 month) for immune, inflammatory, autoimmune diseases, or bacterial/viral infections, (8) blood transfusions in the previous year; (9) celiac disease and a relevant diet modification during the last month. IA and control subjects with previous colonoscopy/sigmoidoscopy with endoscopic resection of a colonic lesion were also excluded.

A total of 620 patients were contacted by the trained interviewers. Of these, around 25% did not meet the eligibility criteria and less than 2% refused to participate in the study. Furthermore, 49 subjects were excluded due to some inaccuracies in the enrolment procedures and 42 due to previous cancers or to other ineligible conditions that were discovered after further data check. From the remaining 347 patients, the final sample after matching included 300 subjects: 100 CRC cases, 100 IA patients and 100 healthy controls.

Colonoscopy and histological examinations were revised by two pathologists who determined CRC cases and their clinical characteristics (e.g., stage, lymph node), as well as IA and their major features (e.g., morphology, measure), and healthy controls.

Cases included 100 incident, histologically confirmed CRC: 62 men and 38 women (mean age: 67, range 31–85 years). Of these, 21 were in the right colon (International Classification of Diseases, 10th Edition, ICD-10, C18.0, C18.2, C18.3), 12 in the transverse colon, in the splenic flexure, and in the descending colon (ICD-10, C18.4, C18.5, C18.6), 17 in the sigmoid colon (ICD-10, C18.7), and 50 in the rectum, including the rectosigmoid junction (ICD-10, C20, C19.9). According to the TNM system, 2 CRC were stage I, 49 stage II, 26 stage III, and 19 stage IV (4 had missing information).

One hundred IA patients (mean age: 66, range 34–84 years) and 100 healthy controls (mean age: 66, range 26–85 years) were included.

The study protocol was revised and approved by the ethical committees of the hospitals involved in data recruitment: ASST Grande Ospedale Metropolitano Niguarda (No. 477-112016) and Fondazione IRCCS Ca’ Granda Ospedale Maggiore Policlinico (No. 742-2017).

### 4.1. Interview

After written consent, a face-to-face interview was performed. The questionnaire included information on socio-demographics, smoking habits, physical activity, anthropometric measures, occupational exposures, medical history, selected drug and supplement use, family history of cancer, sleeping habits and dental care. A food frequency questionnaire was used to assess the past patients’ usual diet.

### 4.2. Blood Collection

Blood samples were collected before the colonoscopy in order to avoid possible bacteria contamination after colonoscope insertion and to keep the same setting for each participant. An aliquot of 7 mL of blood was collected in a tube with EDTA and an aliquot of 3 mL in a blank (without anticoagulant) tube. Three microvials of 1 mL from EDTA tube were immediately stored at –S180 °C for the microbiomic analysis. The remaining blood was processed and centrifuged, and then stored at −80 °C. At the end of data recruitment, blood samples were sent to Vaiomer SAS, Labège, France, for the analysis of the microbiome. The operators were blind to the group assignment and the samples were analyzed in the same experiment, with the same reagent batches and manipulator, in order to keep the signal to noise ratio optimal and to reduce technical variability.

### 4.3. DNA Extraction, qPCR Experiments and Sequencing of 16S rRNA Gene Amplicons

Bacterial DNA quantification and sequencing reactions were performed by Vaiomer SAS using an optimized blood-specific technique [17,54,55]. DNA was extracted from 0.25 mL of whole blood and collected in a final 50 μL extraction volume. Real-time polymerase chain reaction (PCR) amplification was performed using panbacterial primers EUBF 5′-TCCTACGGGAGGCAGCAGT-3′ and EUBR 5′-GGACTACCAGGGTATCTAATCCTGTT-3′ [56], which target the V3-V4 hypervariable regions of the bacterial 16S rRNA gene with 100% specificity (i.e., no eukaryotic, mitochondrial, or Archaea DNA is targeted) and high sensitivity (16S rRNA of more than 95% of bacteria in Ribosomal Database Project database are amplified). The abundance of the 16S rRNA gene in blood samples was measured by qPCR in triplicate and normalized using a plasmid-based standard range. The results were reported as number of copies of 16S rRNA gene per µL of blood. DNA from whole blood was also used for 16S rRNA gene taxonomic profiling applying MiSeq Illumina technology (2 × 300 paired-end MiSeq kit V3). The samples 20,056, 10,251, 10,248 and 20,086 (referring to 2 IA and 2 control subjects) were excluded from the diversity analyses as they did not reach the threshold of 5000 reads.

Then, sequences were analyzed using Vaiomer bioinformatic pipeline to determine bacterial community profiles. Briefly, after demultiplexing of the bar-coded Illumina paired reads, single read sequences were trimmed (reads R1 and R2 to 290 and 240 bases, respectively) and paired for each sample independently into longer fragments, non-specific amplicons were removed and remaining sequences were clustered into OTUs using FROGS v1.4.0 [57] with default parameters. A taxonomic assignment was performed by Blast+ v2.2.30 against the Silva 132 Parc database. OTUs were clustered based on 97% sequence similarities by two steps through swarm algorithm v2.1.6. The first step consisted of a clustering with an aggregation distance equal to 1. The second step consisted of a clustering with an aggregation distance equal to 3. OTUs with relative abundance lower than 0.005% of the whole dataset of reads were removed. All the reads are publicly available in the European Nucleotide Archive (ENA) with the accession number: PRJEB46474.

### 4.4. Bacterial DNA Contamination Assessment

To assess the potential bacterial DNA contamination from environment and reagents, several negative controls were included in the analyses (Appendix A) showing that background noise and blood contamination did not impact the results of this study.

### 4.5. Statistical Analyses

Two-tailed Wilcoxon signed-rank tests and Friedman tests were used to compare 16S rRNA gene between groups. Since 16S rRNA gene distribution was very similar in control and IA subjects, we grouped the tumor-free subjects as comparison group and increased the study power. ORs of CRC and their corresponding 95% CI were estimated through logistic regression models conditioned on the matching variable. The number of 16S rRNA gene copies was included in the models as quintiles (categorically) based on the distribution of controls and IAs, and as continuous variables, with the measurement unit sets to the difference between the upper cut-points of the 4th and 1st quintile (equal to 4328). Multinomial logistic regression was used to estimate separate ORs for colon and rectal cancer and to test for heterogeneity between the two sites.

Analysis on alpha-diversity and beta-diversity indices, as well as on taxonomic variables were computed among 296 subjects (because of 4 missing data due to technical reasons described above).

To assess samples diversity in terms of richness and evenness, various alpha-diversity indices, including Observed, Chao1, Shannon, Simpson and InvSimpson, were calculated by R PhyloSeq v1.14.0 package. Two-tailed Mann–Whitney tests were used to determine differences in terms of alpha-diversity between groups.

To estimate beta-diversity, Permutational Multivariate Analysis of Variance Suing Distance Matrices (PERMANOVA) was applied based on the UniFrac distances, and Principal Coordinates Analysis (PCoA) was applied to visualize possible differences between groups.

Differences in terms of bacterial taxa and OTUs were evaluated through Welch test after DESeq2 normalization of data, based on negative binomial distribution (R package “DESeq2” v1.26.0).

For each statistical analysis, a post-hoc *p*-value adjustment was performed using Hochberg–Benjamini correction, when appropriate.

Random Forest (R libraries: “randomforest”, “caret”, “Boruta”) was used to infer whether there was a set of variables able to discriminate which group the samples belonged to, by the use of training and test sets randomly selected from our samples. In order to decrease the background noise due to the different library size of the samples sequenced, data were normalized: the relative abundance of taxa was multiplied by 16S rRNA gene abundance (determined by qPCR).

## 5. Conclusions

Data confirm the presence of bacterial DNA in blood in healthy adults and indicate that colon cancer patients had a higher DNA bacterial load as compared to healthy, adenoma and rectal cancer subjects. This result supports the hypothesis of a higher passage of bacteria from gastrointestinal tract to bloodstream, with increased community diversity, in colon cancer patients but not in rectal cancer and IA patients.

Moreover, this study found a set of bacterial taxa and OTUs able to discriminate CRC from IA and healthy subjects with an accuracy of 0.70. The confirmation and suitability of this finding in non-invasive diagnostic tests for colon cancer control should be evaluated in larger studies.

## Figures and Tables

**Figure 1 cancers-13-06363-f001:**
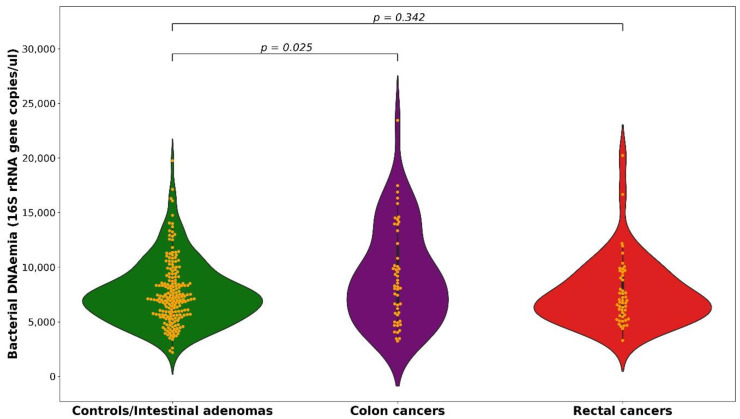
Distribution of 16S rRNA gene copies per µL of whole blood among controls/intestinal adenomas, colon and rectal cancers.

**Figure 2 cancers-13-06363-f002:**
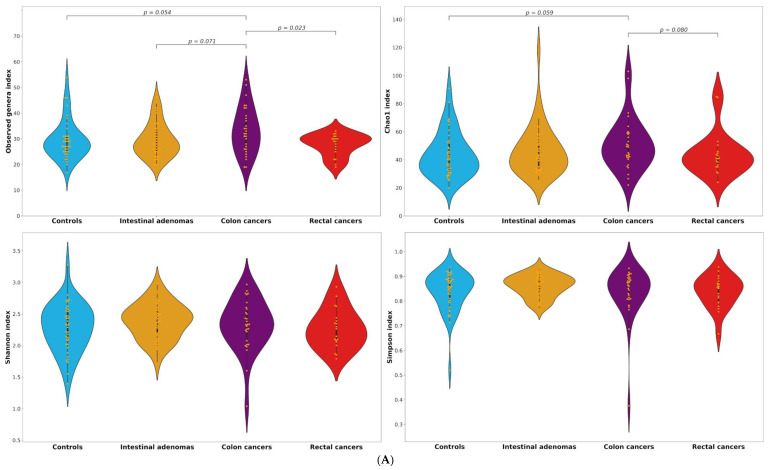
**(A)** Distributions of observed, Chao1, Shannon and Simpson alpha-diversity indices among controls, intestinal adenomas, colon and rectal cancers for genera among the highest two quintiles of 16S rRNA gene copies. (**B**) Distributions of observed, Chao1, Shannon and Simpson alpha-diversity indices among controls, intestinal adenomas, colon and rectal cancers for the operational taxonomic units (OTUs) among the highest two quintiles of 16S rRNA gene copies.

**Figure 3 cancers-13-06363-f003:**
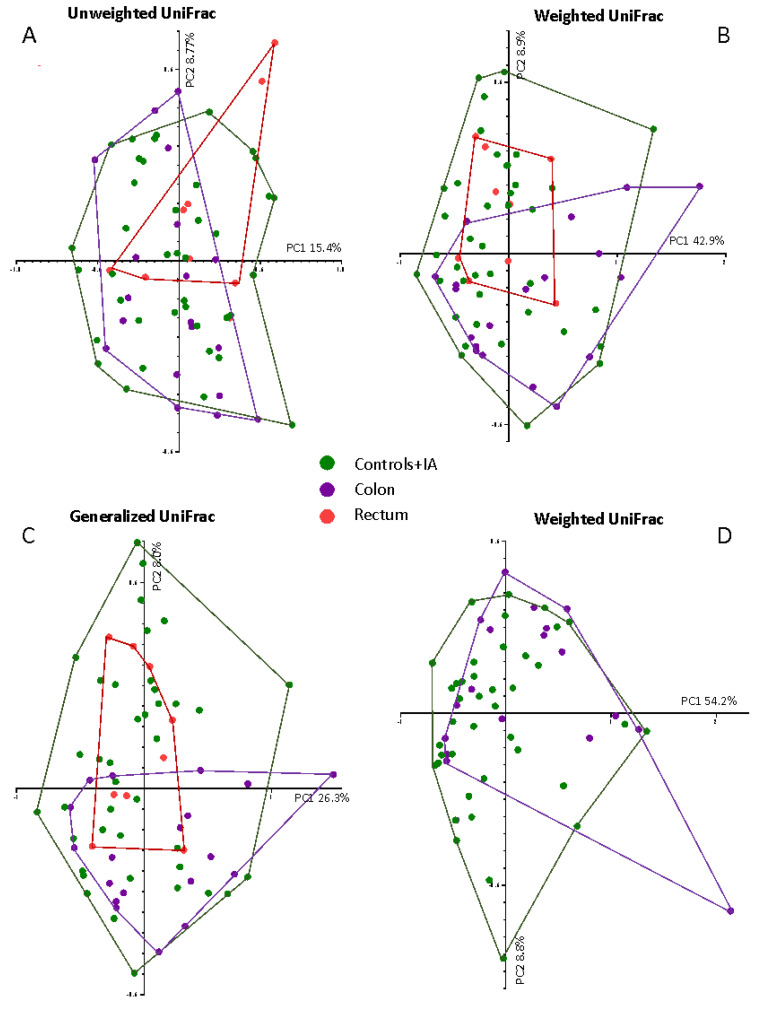
β-diversity of controls and intestinal adenoma (Controls + IA) group, colon cancer (Colon) and rectal cancer (Rectum) among subjects into the fifth quintile of 16S rRNA gene copies. The figure shows the UniFrac ^®^-diversity in all the three variants: (**A**) Unweighted UniFrac; (**B**) Weighted UniFrac; (**C**) generalized UniFrac for all the three groups: Controls + IA, Colon and Rectum. Panel (**D**) shows the Weighted Unifrac for Controls + IA and colon cancers after post-hoc analyses.

**Figure 4 cancers-13-06363-f004:**
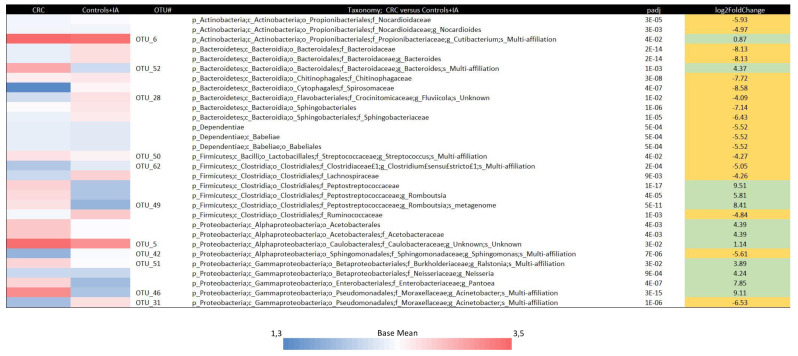
Different taxa between colorectal cancer (CRC) cases and controls and intestinal adenomas (IA) group by DESeq2 analyses. The taxonomic lineage of each taxon is shown: p, phylum; c, class; o, order; f, family; g, genus; OTU#, Operational Taxonomic Unit. The first two columns show the logarithmic transformation of normalized base mean value for each group. The “padj” column shows the p-value for heterogeneity between groups adjusted for multi-testing analyses. Positive fold changes (shown on a green background) designate taxon overrepresentation in the CRC group. Negative fold changes (shown on a yellow background) designate taxon underrepresentation in the CRC group.

**Figure 5 cancers-13-06363-f005:**
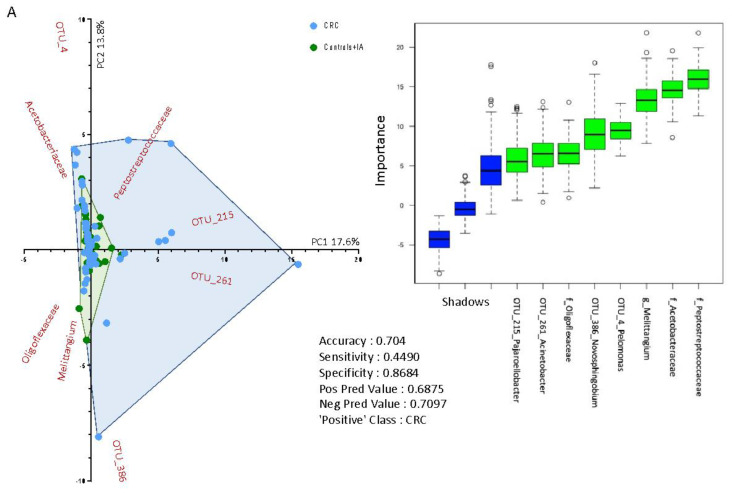
Biplot of predictive variables discriminating (**A**) colorectal cancer (CRC) versus control and intestinal adenoma (IA) subjects; (**B**) colon versus rectal cancer cases, using Random Forest algorithm. The boxplot on the right side of each figure shows the importance (based on mean accuracy level) of the variables by Boruta feature selection. The shadows are part of the Boruta algorithm and show the max, medium and lowest level of mean accuracy, using the same dataset with group labels shuffled. The table in the middle part of each figure shows the Random Forest results considering the ‘Positive’ Class as indicated.

**Table 1 cancers-13-06363-t001:** Distribution of 100 healthy controls, 100 intestinal adenoma (IA) patients and 100 colorectal cancer (CRC) cases by sex, age, study center and years of education. Italy 2017–2019.

Characteristic	Controls	IA	CRC
Sex			
Male	62 (62%)	62 (62%)	62 (62%)
Female	38 (38%)	38 (38%)	38 (38%)
Age group (years)			
<50	7 (7%)	4 (4%)	10 (10%)
50–59	23 (23%)	20 (20%)	19 (19%)
60–69	26 (26%)	36 (36%)	29 (29%)
70–79	33 (33%)	29 (29%)	31 (42%)
≥80	11 (11%)	11 (11%)	11 (11%)
χ^2^ test, *p* = 0.76			
Mean (SD) age (years) *	66.0 (11.8)	65.9 (10.9)	66.1 (11.6)
Center			
Niguarda	65 (65%)	65 (65%)	65 (65%)
Policlinico	35 (35%)	35 (35%)	35 (35%)
Education (years) ^†^			
<7	12 (12%)	19 (19%)	25 (25%)
7–11	24 (24%)	26 (26%)	25 (25%)
≥12	64 (64%)	55 (55%)	49 (50%)
χ^2^ test, *p* = 0.155			

SD: standard deviation (SD). * Anova test for heterogeneity *p* = 1.00. ^†^ The sum does not add up to the total because of one missing value.

**Table 2 cancers-13-06363-t002:** Odds ratios (OR) * and corresponding 95% confidence intervals (CI) according to quintiles of 16S rRNA gene copies in whole blood/µL of 100 control and 100 intestinal adenoma (IA) patients and of 100 colorectal cancer (CRC) cases (50 colon and 50 rectal cancers). Italy 2017–2019.

	Mean (SD)	Quintile of Number of Gene Copies ^†^, OR (95% CI)			χ^2^ Trend (*p* Value) across the 3 Categories	Continuous OR ^§^
		1–3 ^‡^	4	5		
Upper cutpoints (n copies/µL)		7617.5	9707.4	*-*		
Control/IA, *n* (%)	7606.6 (3895.8)	120 (60%)	40 (20%)	40 (20%)		
Total CRC, *n* (%)	8387.1 (2865.4)	52 (52%)	20 (20%)	28 (28%)		
		1 ^‡^	1.16	1.59	2.40	1.39
			(0.60–2.22)	(0.89–2.82)	(0.121)	(1.00–1.92)
Colon cancer, *n* (%)	9145.4 (4476.2)	21 (42%)	10 (20%)	19 (38%)		
		1 ^‡^	1.96	2.62	6.21	2.02
			(0.75–5.08)	(1.22–5.65)	(0.013)	(1.26–3.25)
Rectal cancer, *n* (%)	7628.8 (3075.0)	31 (62%)	10 (20%)	9 (18%)		
		1 ^‡^	0.73	0.81	0.358	0.86
			(0.29–1.84)	(0.32–2.03)	(0.549)	(0.51–1.42)
χ^2^ interaction (*p* value) between colon and rectum				5.30 (0.021)		4.34 (0.037)

SD: standard deviation (SD); * Estimated from logistic regression models, conditioned on age, sex, and study center; ^†^ Computed among control and IA distribution; ^‡^ Reference category; ^§^ Estimated for an increment equal to the difference between the upper cut-points of 4th and the 1st quintiles (=4328 copies).

**Table 3 cancers-13-06363-t003:** Distribution of control and intestinal adenoma (IA) subjects, and colon and rectal cancer cases by cancer subsites and quintiles of 16S rRNA gene copies. Italy 2017–2019.

	Total	Quintile of Number of Gene Copies *, *n* (%)		
		1–3	4	5
Control/IA	200	120 (60%)	40 (20%)	40 (20%)
Tumor subsite				
Right colon	21	7 (33%)	3 (14%)	11 (53%)
Cecum	4	2 (50%)	0 (0%)	2 (50%)
Ascending	11	2 (18%)	2 (18%)	7 (64%)
Hepatic flexure	6	3 (50%)	1 (17%)	2 (33%)
Other than right colon	29	14 (48%)	7 (24%)	8 (38%)
Transverse colon	2	1 (50%)	1 (50%)	0 (0%)
Splenic flexure	3	1 (33%)	1 (33%)	1 (33%)
Descending colon	7	4 (57%)	2 (29%)	1 (14%)
Sigmoid colon	17	8 (47%)	3 (18%)	6 (35%)
Rectum	50	31 (62%)	10 (20%)	9 (18%)
Rectosigmoid junction	3	3 (100%)	0 (0%)	0 (0%)
Rectum	47	28 (60%)	10 (21%)	9 (19%)

* Computed among control and IA distribution.

**Table 4 cancers-13-06363-t004:** Distributions of observed, Chao1, Shannon and Simpson alpha-diversity indices of control, intestinal adenoma (IA), colon and rectal cancer subjects for bacterial genera and OTUs among the highest two quintiles of 16S rRNA gene copies. Italy 2017–2019.

		Median (I–III Quartiles)				*p* * Controls vs IA	*p* * Colon Cancers vs IA	*p* * Colon Cancer vs Controls	*p* * Rectal Cancer vs IA	*p* * Rectal Cancer vs controls	*p* * Colon Cancer vs Rectal Cancer
	*Alpha-Diversity*	Controls	IA	Colon Cancer	Rectal Cancer						
**Genera**	*Observed*	28 (25–31)	28 (25–33.5)	32 (26–39)	29 (25–31)	0.942	0.071	0.054	0.714	0.968	0.023
	*Chao1*	40.6 (31.7–52.3)	42.5 (33.4–52.5)	49 (41.5–59)	41 (35–46)	0.476	0.148	0.059	0.703	0.789	0.080
	*Shannon*	2.33 (2,05–2,55)	2.41 (2.18–2.56)	2.33 (2.08–2.58)	2.26 (2.06–2.44)	0.254	0.535	0.614	0.119	0.715	0.442
	*Simpson*	0.86 (0.80–0.89)	0.87 (0.84–0.89)	0.84 (0.81–0.89)	0.86 (0.79–0.88)	0.394	0.496	0.846	0.186	0.697	0.513
**OTUs**	*Observed*	34 (30–39)	35 (32.4–43.5)	40 (33–51)	37 (30–38)	0.413	0.154	0.039	0.570	0.820	0.029
	*Chao1*	53.4 (47.8–70.5)	66 (51.9–92.8)	71.1 (52–87)	56 (51–73)	0.070	0.981	0.067	0.233	0.565	0.278
	*Shannon*	2.52 (2.27–2.74)	2.63 (2.46–2.84)	2.60 (2.46–2.73)	2.50 (2.27–2.65)	0.154	0.662	0.473	0.089	0.727	0.149
	*Simpson*	0.90 (0.86–0.93)	0.91 (0.88–0.93)	0.90 (0.87–0.92)	0.87 (0.86–0.91)	0.233	0.473	0.653	0.062	0.403	0.229

* *p* for heterogeneity estimated from the Wilcoxon rank-sum.

## Data Availability

All the reads are publicly available in the European Nucleotide Archive (ENA) with the accession number: PRJEB46474.

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
