# Peer review of "Blood Bacterial DNA Load and Profiling Differ in Colorectal Cancer Patients Compared to Tumor-Free Controls"

_cancers, 2021, doi:10.3390/cancers13246363_

Round 1

Reviewer 1 Report

I read an article entitled "Blood bacterial DNA load and profiling differ in colorectal cancer patients compared to tumor-free controls" with great interest. This well-designed and well-written study, I will add just a few to improve the work.

1. What is your criteria for selecting an IA group?
2. Which method/s of strategy are used for validation of data?
3. What is the scientific logic for merging the control group and the polyp group? Is the logic stated in the article sufficient?
4. What are the pathological stages of the case group ?
5. What is the scientific reason for the difference between the variables (blood 16S rRNA gene copies) in colon and rectal cancer when it is a blood sample? Shouldn't this issue be further explored in the discussion section?
6. What is the scientific reason for the difference between the variables in different tumor locations when it is a blood sample? Shouldn't this issue be further explored in the discussion section?
7. What are the limitations of this study?

Author Response

  1. What is your criteria for selecting an IA group?

Response:

We included IA adenoma group because IA is a preneoplastic condition sharing various etiological factors with CRC, as described in the Introduction. As it was described in the Discussion, “the inclusion of IA allowed to investigate an important phase on the mechanisms behind the process of the adenoma-carcinoma sequence”. An open question was to evaluate whether the inflammation status, which is already present in IA, could influence intestinal barrier permeability and consequently modify the translocation, even before the process of carcinogenesis. However, we did not find differences in terms blood bacterial DNA in IA patients as compared to healthy subjects. Thus, we better specified this in the text (Conclusions, page 20) indicating that our data did not necessarily support the hypothesis of a higher passage of bacteria from gastrointestinal tract to bloodstream in IA patients compared to healthy subjects.

  1. Which method/s of strategy are used for validation of data?

Response:

As we specified in the Methods and Supplementary material, negative controls consisting of ultrapure water were added at the DNA extraction step in order to assess the potential bacterial DNA contamination from environment and reagents and to validate metagenomic data.

Moreover, we specified in the Methods (page 19) that, when we applied Random Forest algorithms, data were randomly separated into training and test sets in order to validate the results. However, “the lack of an ad hoc developed validation cohort remains a limitation of our work.”, as we added in the Discussion (page 16).

  1. What is the scientific logic for merging the control group and the polyp group? Is the logic stated in the article sufficient?

Response:

As indicated in the Response 1, we first considered IA as an outcome variable and found no differences in terms of blood bacterial DNA load as compared to healthy subjects (p for heterogeneity=0.95). On the basis of this result, in order to increase the study power, we merged control and IA groups as a single comparison group. We added in the Methods (pages 18-19) that “since 16S rRNA gene distribution was very similar in control and IA subjects, we grouped the tumor-free subjects as comparison group and increased the study power”. We then compared their 16S rRNA gene copy distribution with that of CRC, colon and rectal cancer cases.

  1. What are the pathological stages of the case group ?

Response:

We added in the Methods that “According to the TNM system, 1 CRC was stage I, 49 stage II, 26 stage III, and 19 stage IV (5 had missing information)”.

  1. What is the scientific reason for the difference between the variables (blood 16S rRNA gene copies) in colon and rectal cancer when it is a blood sample? Shouldn't this issue be further explored in the discussion section?

Response:

We further discussed the differences between colon and rectum and the possible reasons for different blood 16S rRNA gene copies and bacterial translocation (Discussion, page 16). Moreover, in the limitations of the study we specified that “It is not easy to explain the lack of difference between rectal cancer and tumor-free subjects, since the bacterial translocation can be different due to the related inflammatory condition in rectal cancer patients. A possible role of size or shape selectivity for bacteria transiting from the intestine to the bloodstream in different locations could explain at least in part this result but a limitation of this study is the lack of fecal samples on our subjects, which not allowed us to further analyze this issue.”

  1. What is the scientific reason for the difference between the variables in different tumor locations when it is a blood sample? Shouldn't this issue be further explored in the discussion section?

Response:

We further described the features that can play a role in changing the extent of bacterial translocation along the lower intestinal tract. We specified that “local differences in underlying genetics, including genetic expression and immunological activity have been highlighted, along with a negative gradient of immune cells from proximal colon to distal colon and rectum. These features can play a role in changing the extent of bacterial translocation along the lower intestinal tract, leading to differences in blood 16S rRNA gene copies according to CRC localization. Moreover, gut microbiota and mechanisms of carcinogenesis also vary from left to right colon and to rectum. Another factor that could affect regional differences in bacterial translocation and blood 16S rRNA gene copies is the organization and thickness of mucus, which is looser and less organised in the proximal colon, facilitating bacterial contact with the epithelium and, thus, translocation”; and we added 2 references: [1,2]. Moreover, results on colon cancer subsites were toned down adding that results “should be interpreted with caution given the reduced number of involved cases”.

  1. Herath, M.; Hosie, S.; Bornstein, J.C.; Franks, A.E.; Hill-Yardin, E.L. The role of the gastrointestinal mucus system in intestinal homeostasis: Implications for neurological disorders. Front Cell Infect Microbiol 2020, 10, 248.
  2. Kamphuis, J.B.J.; Mercier-Bonin, M.; Eutamene, H.; Theodorou, V. Mucus organisation is shaped by colonic content; a new view. Sci Rep 2017, 7, 8527.

  1. What are the limitations of this study?

Response:

As suggested by the reviewer 1, we expanded the section of limitations in the Discussion (page 16). Among others, we specified that “the lack of an ad hoc developed validation cohort remains a limitation of our work” and that one of the limitations of this study is the lack of fecal samples on our subjects, which not allowed us to further analyze intestinal microbiota in our analyses.

Reviewer 2 Report

The manuscript entitled “Blood bacterial DNA load and profiling differ in colorectal cancer patients compared to tumour-free controls by Mutignani M. et al., evaluates the possible role of the bacterial intestinal microbiome on colorectal cancer risk. The work is based on the study of bacterial ribosomal 16S profiling in blood samples. thus on bacteria that translocate from the intestinal tract to the blood, in a clinical define context. 

These data implicate an almost equal dysfunction of the intestinal epithelial barrier permeability for patients with cancer or without cancer. This is, according to this referee, the weak point of the study. Indeed, the type of bacteria (by their shape, motility, capture by IgA etc…) that escape from the intestine to the blood through epithelial junctions can be very different in a non-inflammatory condition (tumour-free) versus a high inflammatory context that is expected with cancer (CRC). The authors report that the profile of 16S bacterial ribosomal was highly modified only in colon cancer and not in rectal cancer. The rectal epithelium, however, could be not identical to the colon epithelium in terms of junction opening dynamics and, of course, of bacterial translocation. On this point, the work appears not totally convincing.

The manuscript is well written and supported by a significant amount of data but the problem of size or shape selectivity for bacteria transiting from the intestine to the bloodstream in different conditions is a critical point that should be faced or at least, critically discussed.

Author Response

We agreed with comments of the reviewer 2 and discussed the main issues raised by the reviewer. In particular, we added that “Our data show a non trivial dysfunction of the intestinal epithelial barrier permeability also for patients without cancer, but this is in line with previous studies that characterized blood microbiota among healthy subjects. However, it is not easy to explain the lack of difference between rectal cancer and tumor-free subjects, since the bacterial translocation can be different due to the related inflammatory condition in rectal cancer patients. A possible role of size or shape selectivity for bacteria transiting from the intestine to the bloodstream in different locations could explain at least in part this result but a limitation of this study is the lack of fecal samples on our subjects, which not allowed us to further analyze this issue.” We added this reference [3]

  1. Genua, F.; Raghunathan, V.; Jenab, M.; Gallagher, W.M.; Hughes, D.J. The role of gut barrier dysfunction and microbiome dysbiosis in colorectal cancer development. Front Oncol 2021, 11, 626349.

Moreover, we added that “the lack of an ad hoc developed validation cohort remains a limitation of our work”

Reviewer 3 Report

CRC (colorectal cancer) is the 3 rd more common cancer worldwide with an important moratlity and morbidity. CRC derives from a sequential accumulation of genetic alterations that involves the transition from normal mucosa to pre-malignant lesions with progression to intestinal adenoma (IA) and invasive CRC. Some mechanism of carcinogenessis are analyzed such inflammation and immunity that are inextricably linked to all phase of CRC development and are asocciated with dysfunction of gut mucosal barrier. There are some studies that analyzed the intestinal microbial ecosystem influences of IA and CRC patients and demonstrated that epithelial barrier dysfunction can lead to increased intestinal permeability, and a greater bacterial translocation from gastrointestinal tract to blood stream in IA and CRC.

This current article summarizes the role of bacterial translocation from gastrointestinal tract to blood stream and it’s association with intestinal adenoma and/or colorectal cancer risk, using an epidemiological and metagenomic approach to evaluated the relation of the bacterial DNA load and the bacterial taxonomic groups assed by 16S rRNA profiling.

In your article you presented very well that colon cancer patients have an overrepresentation of bacterial DNA in blood as compared to tumor-free controls, including intestinal adenoma or healthy subjects. You pointed that topography of CRC cancer is an important criteria, and results appeared stronger for cancers in the right colon, whereas no difference in terms of bacterial load was found for rectal cancer. CRC patients had an enrichment of Peptostreptococcaceae and Acetobactriaceae and a reduction of Bacteroidaceae, Lachnospiraceae and Ruminococcaceae. Bacteria have shown capacity to interact directly with immune system cells and to impact in multiple host functions, also can disseminate to liver through the disruption of gut vascular barrier in colorectal cancer with hepatic metastasis.An important role in bacterial translocation may have the mucus layer, which were found to vary along the colon. Your results can be relevant for the detection of right colon cancers, which often have a subtle presentation and more advanced stage at diagnosis.

This present article is written in a clear and concise manner and highlighted that DNA bacterial load and different bacterial profiling have crucial role in the carcinogenesis in patients with CRC compare with IA or healthy subjects, with precision and clarity, proving to be a launch platform in it’s field. It is important to support the need of further studies in order to reach a definitive conclusion.

Author Response

We agreed with the reviewer comments and stressed that it is important to support the need of further studies. We added in the Discussion (page 17) that “the lack of an ad hoc developed validation cohort remains a limitation of our work”.

We remain at your disposal if further information or clarifications are needed.

Round 2

Reviewer 2 Report

The Authors addressed the criticism. The manuscript is acceptable in the present form